# Dorsal hippocampus mediates light–tone associations in male mice

**Julia S Pinho†, Carla Ramon-Duaso, Irene Manzanares-Sierra, Arnau Busquets-Garcia***

Cell-Type Mechanisms in Normal and Pathological Behavior Research Group, Neuroscience Research Program, Hospital del Mar Research Institute, Barcelona, Spain

## eLife Assessment

Pinho et al use in vivo calcium imaging and chemogenetic approaches to examine the involvement of hippocampal sub-regions across the different stages of a sensory preconditioning task in mice. They find evidence for sensory preconditioning in male mice. They also find that, in these mice, CaMKII-positive neurons in the dorsal hippocampus encode the audio-visual association that forms in stage 1 of the task. The evidence in support of these findings is **convincing**. The **important** study will be of interest to researchers in the fields of learning and memory and/or hippocampus function.

*For correspondence:
abusquets@researchmar.net

Present address: †Gulbenkian Institute for Molecular Medicine, Oeiras, Portugal

Competing interest: The authors declare that no competing interests exist.

**Abstract** Daily choices are often influenced by environmental cues that are not directly associated with reinforcers. This phenomenon, known as higher-order conditioning, can be studied using sensory preconditioning tasks in rodents. This behavioral paradigm involves the repeated pairing of two innocuous stimuli, such as a light and a tone, followed by a devaluation phase in which one stimulus is associated with an unconditioned stimulus, such as a mild footshock. The result is a conditioned response (e.g., freezing) to both the conditioned stimulus (direct learning) and the non-conditioned stimulus (mediated learning). In our study, we successfully established a light–tone sensory preconditioning task specifically in male mice, as sex differences were observed in both control experimental groups and in sensory preconditioning responses. We employed in vivo, freely moving fiber photometry to monitor neural activity in the dorsal and ventral subregions of the hippocampus in male mice during the formation of associations between innocuous stimuli and reinforcers. Additionally, we combined our sensory preconditioning task with chemogenetic approaches to investigate the roles of these hippocampal subregions in sensory preconditioning. Our results indicate that dorsal, but not ventral, CaMKII-positive neurons are involved in encoding innocuous stimuli during the preconditioning phase. Overall, we developed a novel light–tone sensory preconditioning protocol in male mice, enabling the detection of sex differences and furthering our understanding of how specific hippocampal subregions and cell types regulate complex cognitive processes.

## Introduction

The brain mechanisms underlying associative learning have traditionally been elucidated through classical conditioning paradigms involving salient stimuli as reinforcers—such as pairing an electric footshock with innocuous stimuli (e.g., a light, tone, or odor). However, these paradigms are not sufficient to fully capture the complexity of animal behavior, as daily choices are not always dictated by stimuli that have been directly associated with a potent reinforcer. In both humans and animals, responses often occur to cues that were never explicitly conditioned but had previously been associated with

other stimuli linked to specific aversive or rewarding outcomes (*Wimmer and Shohamy, 2012*; *Gewirtz and Davis, 2000*; *Parkes and Westbrook, 2011*; *Holmes et al., 2022*; *Ioannidou et al., 2021*).

These higher-order conditioning processes can be studied in laboratory settings using sensory preconditioning procedures (*Gewirtz and Davis, 2000*; *Holmes et al., 2013*; *Parkes and Westbrook, 2010*; *Robinson et al., 2014*; *Gostolupce et al., 2022*; *Hart et al., 2020*; *Busquets-Garcia et al., 2018*). These tasks involve incidental associations between neutral stimuli (e.g., odors, tastes, lights, or tones) during a preconditioning phase, followed by classical conditioning of one of these stimuli with an aversive or appetitive unconditioned reinforcer (*Gewirtz and Davis, 2000*; *Holmes et al., 2013*; *Parkes and Westbrook, 2010*; *Robinson et al., 2014*; *Gostolupce et al., 2022*; *Hart et al., 2020*; *Busquets-Garcia et al., 2018*). As a result, subjects exhibit aversion or preference toward the stimulus that was never directly paired with the reinforcer, enabling the assessment of mediated learning (*Gewirtz and Davis, 2000*; *Holmes et al., 2013*; *Parkes and Westbrook, 2010*; *Robinson et al., 2014*; *Gostolupce et al., 2022*; *Hart et al., 2020*; *Busquets-Garcia et al., 2018*). While the brain circuits underlying classical associative memory between neutral cues and reinforcers have been extensively studied, the mechanisms responsible for incidental associations between innocuous sensory stimuli—and their behavioral consequences—remain much less understood.

The hippocampus and other cortical regions (e.g., the perirhinal, retrosplenial, and orbitofrontal cortices) have been implicated in higher-order conditioned responses (*Holmes et al., 2013*; *Robinson et al., 2014*; *Hart et al., 2020*; *Busquets-Garcia et al., 2018*; *Kahnt and Schoenbaum, 2021*; *Iordanova et al., 2011*; *Wheeler et al., 2013*). Sensory preconditioning tasks using various sensory modalities have shown that the involvement of different brain regions may depend on the behavioral phase being studied (i.e., preconditioning or testing) or the modality of the sensory cues (visual, gustatory, olfactory, or auditory). However, it is still unknown whether there are common brain regions where different types of stimuli are integrated, regardless of the behavioral phase or the modality. The hippocampus has been proposed to play a central role in multiple behavioral phases of sensory preconditioning procedures across species (*Wimmer and Shohamy, 2012*; *Holmes et al., 2022*; *Busquets-Garcia et al., 2018*). This region, which maintains continuous information exchange with other cortical areas, is thought to act as an integrator of past experiences into broader cognitive representations through a tightly regulated excitatory–inhibitory balance (*Caroni, 2015*; *Chevaleyre and Piskorowski, 2014*). Previous studies have highlighted a specific role for hippocampal mechanisms during the encoding of incidental associations (*Busquets-Garcia et al., 2018*). However, the contribution of specific hippocampal subregions or cell types to sensory preconditioning remains unclear. The dorsoventral axis of the rodent hippocampus is known to be structurally and functionally segregated. The dorsal hippocampus, connected to cortical regions and the thalamus, is primarily involved in cognitive processes such as navigation and exploration (*Fanselow and Dong, 2010*; *Strange et al., 2014*; *Jones and Witter, 2007*; *Gálvez-Márquez et al., 2022*; *Moser et al., 1993*). In contrast, the ventral hippocampus, which communicates with the amygdala, nucleus accumbens, and hypothalamus, plays a key role in motivated and emotional behaviors (*Fanselow and Dong, 2010*; *Strange et al., 2014*; *Groenewegen et al., 1987*). Based on this, our initial hypothesis was that the dorsal and/or ventral hippocampus may differentially contribute to sensory preconditioning in mice.

To test this, we employed chemogenetic and imaging techniques in combination with an adapted light–tone sensory preconditioning ($_{LT}$SPC) procedure (*Holmes et al., 2013*). Our results suggest that the dorsal hippocampus, and in particular CaMKII-positive neurons, plays a crucial role in the formation of incidental associations between visual and auditory stimuli, ultimately driving sensory preconditioning responses. Understanding the differential contribution of the dorsal and ventral hippocampus to sensory preconditioning provides valuable insights into the mechanisms of higher-order conditioning and the broader role of the hippocampus in associative learning.

## Results

### Simultaneous light–tone associations during preconditioning are required for sensory preconditioning responding in male mice

Sensory preconditioning tasks using olfactory and gustatory stimuli have already been demonstrated in mice in several previous studies (*Busquets-Garcia et al., 2018*; *Wheeler et al., 2013*). However, other sensory modalities such as visual and auditory stimuli are also highly relevant for animals' daily

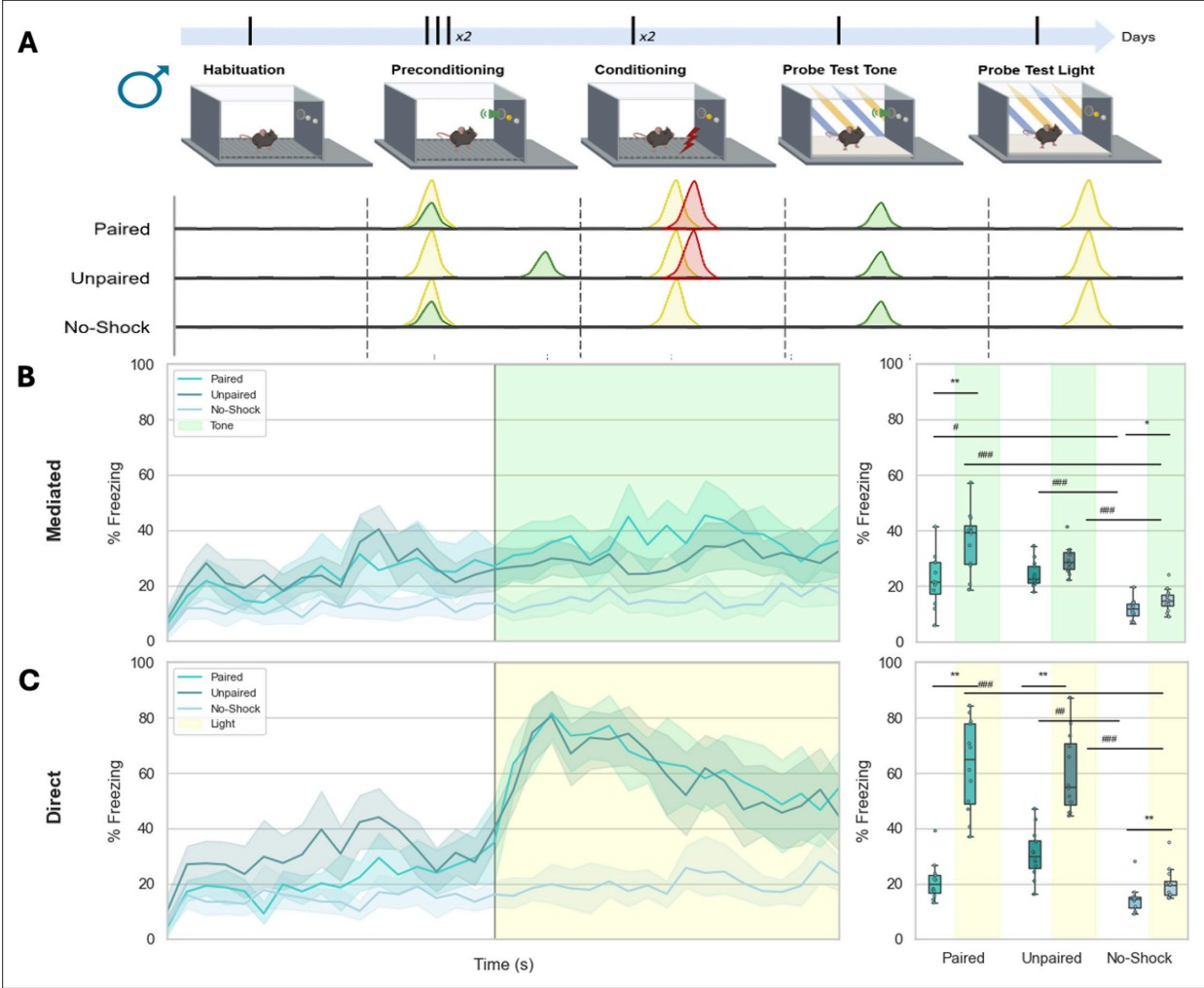

**Figure 1.** Simultaneous associations between light and tone are required for sensory preconditioning responding. (**A**) Schematic representation of the $_{LT}$SPC task in males (two training sessions) with a representation of paired, unpaired, and no-shock experimental groups. The temporal dynamics of freezing represented in bins of 10 s across time of experiment of the Probe Test 1 (tone) (**B** on the left) and Probe Test 2 (light) (**C** on the left). The percentage of time spent freezing during OFF and ON periods of the Probe Test 1 (tone) (**B** on the right) and Probe Test 2 (light) (**C** on the right). *Significant p-value (<0.05) after false discovery rate (FDR). GLM: generalized linear model fitted to gamma distribution with planned comparisons. KW: Kruskal–Wallis across experimental groups: paired, unpaired, no-shock. * p<0,05; ** p<0,01 (between OFF and ON periods); # p<0,05; ## p<0,01, ###, p<0,001. See statistical details in *Supplementary file 1A*.

The online version of this article includes the following figure supplement(s) for figure 1:

**Figure supplement 1.** New automatized tool to measure freezing responses.

**Figure supplement 2.** Simultaneous associations between light and tone to develop a sensory preconditioning protocol in females.

**Figure supplement 3.** Spearman correlations of mediated (tone) and direct (light) learning for the protocol of sensory preconditioning for males (**A**) and females (**B**).

decision-making. While the use of these cues to assess aversive sensory preconditioning has been shown in rats (*Holmes et al., 2013*; *Robinson et al., 2014*; *Iordanova et al., 2011*), their use in mice—particularly for aversive paradigms—remains limited (*Busquets-Garcia et al., 2018*). The first aim of the present work was to establish a sensory preconditioning task in mice using auditory and visual cues, enabling the assessment of both mediated and direct learning. To this end, we developed a sensory preconditioning protocol (see Methods, *Figure 1A*), alongside a novel automated tool to analyze the freezing response (i.e., immobility) in mice. This tool combines DeepLabCut pose estimation with a newly developed Python script, and its accuracy was validated showing a high correlation with manual scoring and ezTrack (*Pennington et al., 2021*) freezing assessments (*Figure 1—figure*

*supplement 1*). Although for freezing scoring, our tool is similar to ezTrack measurements, there are several key added benefits offered by this approach, such as an enhanced flexibility to combine freezing measurements with other customized behavioral assessments, which can be particularly beneficial for labs conducting large-scale or longitudinal behavioral studies in rodents.

Using both male and female mice, we conducted a long-term sensory preconditioning ($_{LT}$SPC) experiment with three experimental groups in male (*Figure 1A*) and female mice (*Figure 1—figure supplement 2A*): Paired group (protocol described in methods section); Unpaired group (same protocol, but the light and tone were never simultaneous); and No-Shock group (same as the Paired group but without electric footshock exposure) (*Figure 1A*, *Figure 1—figure supplement 1A*). According to this design, male (*Figure 1B, C*) and female mice (*Figure 1—figure supplement 2B, C*) in the Paired group exhibited an enhanced freezing response at the onset of the tone or light, respectively. Specifically, exposure to the tone (Probe Test 1, ON period) led to increased freezing behavior (*Figure 1B*) compared to the OFF period, indicating sensory preconditioning responding in male mice. When the light was presented (Probe Test 2), animals showed greater freezing during the ON period compared to the OFF period (*Figure 1C*), revealing direct learning.

When the light and tone were temporally separated (Unpaired group), male mice failed to exhibit sensory preconditioning (*Figure 1B*), while their direct learning response to the light remained intact (*Figure 1C*). In contrast, female mice still displayed a behavioral response to the tone despite the separation of stimuli, which could be attributed to fear generalization (*Figure 1—figure supplement 2B*), along with preserved direct learning (*Figure 1—figure supplement 2C*). These results reveal sex differences in sensory preconditioning performance: male mice clearly demonstrated both mediated and direct learning, whereas female mice did not. To further explore this, we conducted an independent experiment to test whether the fear generalization observed in the female Unpaired group could be reduced by either extending the interval between cue presentations during preconditioning or by removing the light during the conditioning phase (*Figure 1—figure supplement 2D*). Results showed that under both conditions, female mice still exhibited cue-induced fear generalization, which appears to mask their capacity for mediated learning responding.

Finally, in the No-Shock group, neither male (*Figure 1B*) nor female (*Figure 1—figure supplement 2B*) mice showed a stimulus-mediated response, suggesting that tone or light exposure alone during the Probe Tests does not elicit behavioral changes. Altogether, these results confirm the successful establishment of an $_{LT}$SPC protocol in male mice, which can now be used to further investigate the brain circuits involved in higher-order conditioning. However, additional research will be required to optimize an $_{LT}$SPC protocol for female mice that avoids the confounding effect of fear generalization. Indeed, supporting these sex differences in this sensory preconditioning task, in male mice we observed a clear correlation between direct learning and mediated learning responding (*Figure 1—figure supplement 3A*), whereas this is not shown in female mice (*Figure 1—figure supplement 3B*).

## Hippocampal cells are engaged during $_{LT}$SPC

Previous studies using sensory preconditioning paradigms with other sensory modalities have suggested a key role for the hippocampus in processing incidental associations between innocuous stimuli during the preconditioning phase (*Wimmer and Shohamy, 2012*; *Busquets-Garcia et al., 2018*). Specifically, hippocampal GABAergic neurons expressing the type-1 cannabinoid receptors, but not the activity of PV interneurons, have been implicated in odor–taste associations in a mouse odor–taste sensory preconditioning task (*Busquets-Garcia et al., 2018*). However, whether similar cellular mechanisms are involved in sensory preconditioning using different sensory modalities remains unknown. To better characterize the involvement of the dorsal and ventral hippocampus, as well as their specific cell types, during our $_{LT}$SPC task, we performed simultaneous in vivo fiber photometry recordings by implanting an optic fiber in each hippocampal subregion of PV-Cre mice and infusing a mixture of AAV-Syn-RCaMP and AAV-DIO-GCaMP viruses (see *Methods*) in both hippocampal subregions. This allowed us to monitor general neuronal activity (via RCaMP) and the activity of PV-positive interneurons (via GCaMP) during both the preconditioning (light–tone pairings) and the conditioning phase (light–footshock pairings). Four weeks after surgery, animals were subjected to the $_{LT}$SPC task (*Figure 2A*). Neuronal activity in the dorsal and ventral hippocampus was quantified as $\Delta F/F$ following stimulus pairings (dHPC: *Figure 2B* on left, vHPC: *Figure 2C* on left). Peri-event time histograms (PETHs) of *z*-scored $\Delta F/F$ revealed the average dynamic response in each subregion

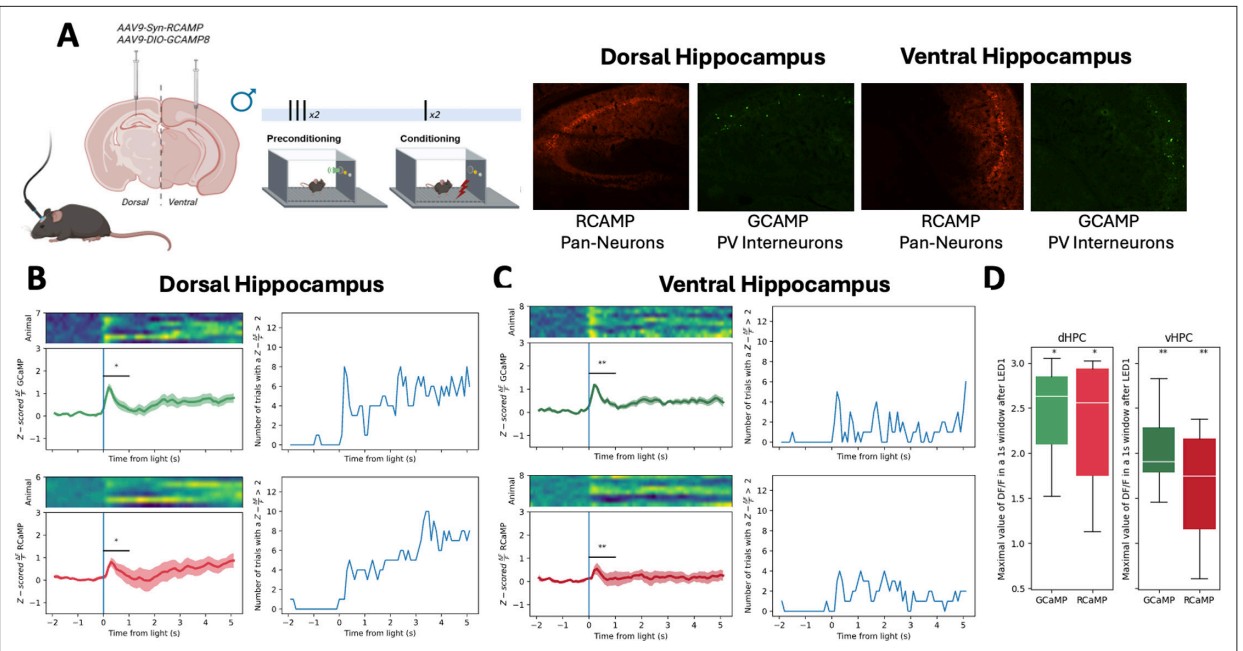

**Figure 2.** Hippocampal cell activity during $_{LT}$SPC. (**A**) Schematic representation of $_{LT}$SPC task while fiber photometry recordings of RCaMP (calcium sensor in synapsin-positive neurons) and Cre-dependent GCaMP (calcium sensor in PV-positive interneurons) in dHPC and vHPC of PV$^{cre}$ mice. (**B**) dHPC modulation during preconditioning of $_{LT}$SPC: on the left upper panel z scores of $\frac{\Delta f}{f}$ (where f represents fluorescence) of GCaMP PV-positive interneurons on dHPC (green), left bottom z scores of $\frac{\Delta f}{f}$ of RCaMP in neurons of dHPC (red), right upper number of events with a $\frac{\Delta f}{f} > 2$ of GCaMP PV-positive interneurons on dHPC, and right bottom number of trials with a $\frac{\Delta f}{f} > 2$ of Rcamp in neurons of dHPC. (**C**) vHPC modulation during $_{LT}$SPC: on left upper panel z scores of $\frac{\Delta f}{f}$ of GCaMP PV-positive interneurons of vHPC (green), left bottom z scores of $\frac{\Delta f}{f}$ of RCaMP in neurons of vHPC (red), right upper number of trials with a $\frac{\Delta f}{f} > 2$ of GCaMP in PV-positive interneurons of vHPC, and right bottom number of trials with a $\frac{\Delta f}{f} > 2$ of RCaMP in neurons of vHPC. (**D**) maximal value of $\frac{\Delta f}{f}$ in the first 1-s window after pairings compared with baseline, on the left (dHPC) and on the right (vHPC). * p-<0.05; ** p<0,01. See statistical details in **Supplementary file 1A**.

The online version of this article includes the following figure supplement(s) for figure 2:

**Figure supplement 1.** Fiber photometry recordings of RCaMP (calcium sensor in synapsin-positive neurons) and Cre-dependent GCaMP (calcium sensor in PV-positive interneurons) in dHPC and vHPC of PVcre mice.

**Figure supplement 2.** Hippocampal cell activity at the onset of the light in conditioning sessions.

**Figure supplement 3.** Hippocampal cell activity at the electric footshock onset.

during light–tone associations (dHPC: *Figure 2B* on right, vHPC: *Figure 2C* on right). These PETHs highlight a lower trial-by-trial variability in dHPC (*Figure 2B*) and a more sustained response over time compared to vHPC (*Figure 2C*). To quantify this activity, we measured the peak z-scored $\Delta F/F$ within a 1-s window following stimulus onset and compared it to baseline values (*Figure 2D*). Notably, this increase in activity was consistent across all preconditioning sessions, with no significant differences observed when data were analyzed per session (*Figure 2—figure supplement 1A*). Additionally, both neuronal activity markers (RCaMP and GCaMP) showed clear increases during the conditioning phase, both at the onset of the light stimulus (conditioned stimulus; *Figure 2—figure supplement 2*) and at the time of footshock delivery (unconditioned stimulus; *Figure 2—figure supplement 3*). These increases were consistent across sessions (*Figure 2—figure supplement 1B, C*), with no significant session-dependent variation. Together, these findings demonstrate the engagement of hippocampal neurons, including PV-positive interneurons, in both the dHPC and vHPC during associative learning in our $_{LT}$SPC task. This supports the role of the hippocampus in processing sensory associations involving non-olfactory modalities.

Although our data showed a clear engagement of hippocampal activity at different phases of our $_{LT}$SPC task, the use of a synapsin promoter is too broad to selectively point to a specific cell type mediating the behavioral responses observed. Thus, we decided to use a similar approach and specifically monitor CaMKII+ neurons during the preconditioning sessions by infusing an AAV-CaMKII-GCAMP6 (see Methods, *Figure 3A*). Interestingly, and matching what was observed with a pan promoter (i.e.,

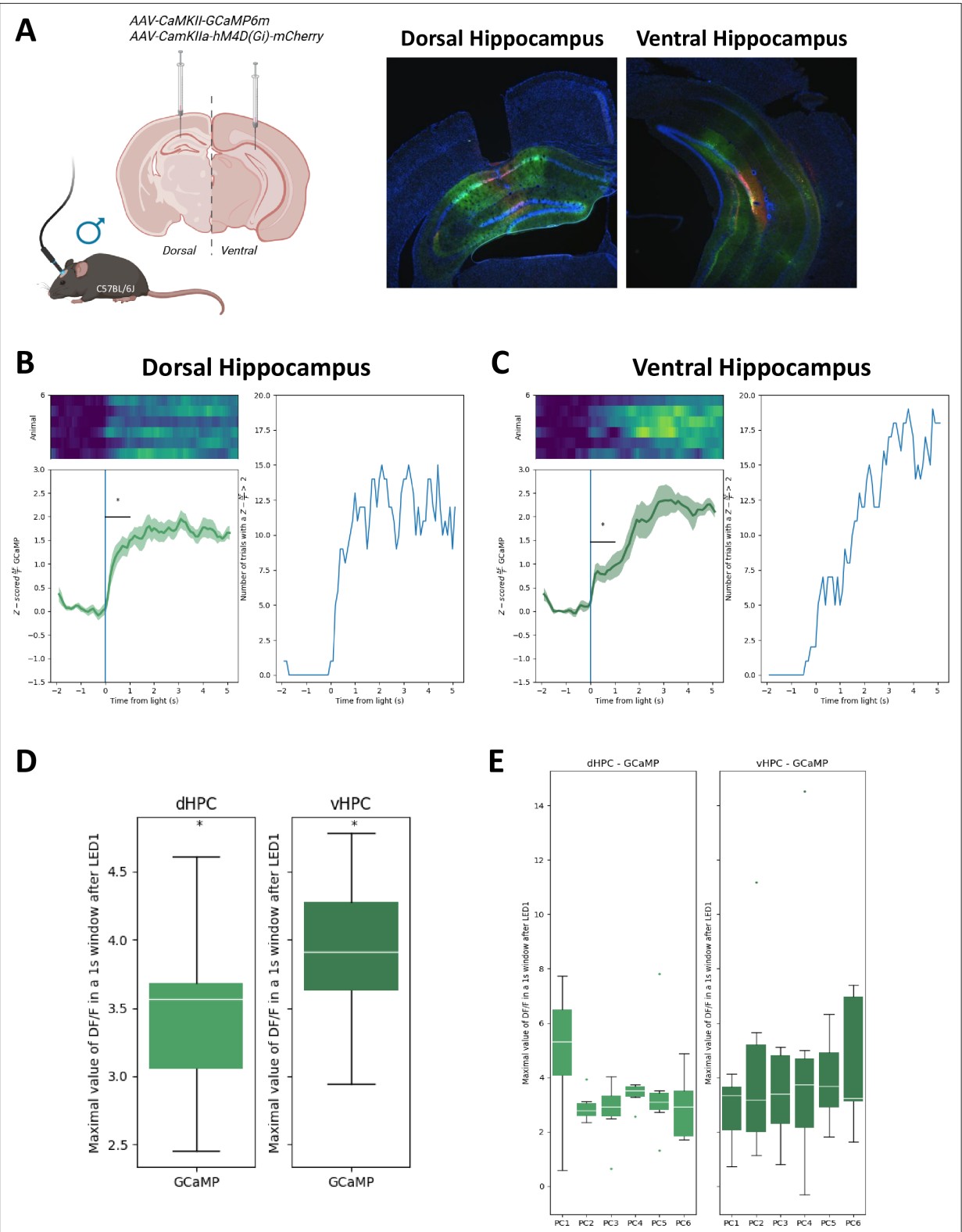

**Figure 3.** CaMKII neurons during preconditioning on dHPC and vHPC. (**A**) Maximal value of $\frac{\Delta f}{f}$ in the first 1-s window after stimulus compared with baseline, for each stimulus during preconditioning session, on the left (dHPC) and on the right (vHPC). (**B**) dHPC modulation during preconditioning of $_{LT}$SPC: on left $z$ scores of $\frac{\Delta f}{f}$ (where $f$ represents fluorescence) of GCaMP in CaMKII-positive neurons on dHPC (green), on right number of events with a $\frac{\Delta f}{f} > 2$ of GCaMP CaMKII-positive neurons on dHPC. (**C**) vHPC modulation during preconditioning of $_{LT}$SPC: on left $z$ scores of $\frac{\Delta f}{f}$ (where $f$ represents fluorescence) of GCaMP in CaMKII-positive neurons on vHPC (green), on right number of events with a $\frac{\Delta f}{f} > 2$ of GCaMP CaMKII-positive

*Figure 3 continued*

neurons on vHPC. (**D**) Maximal value of $\frac{\Delta f}{f}$ in the first 1-s window after stimulus compared with baseline, for the average of the six pairings stimuli during preconditioning session, on the left (dHPC) and on the right (vHPC).(**E**) Photometry recording during each preconditioning trial. * p<0.05. Statistical details in *Supplementary file 1A*.

synapsin), PETHs of *z*-scored Δ*F/F* (**Figure 3B, C**) and the subsequent histograms representing the peak *z*-scored Δ*F/F* within a 1-s window following stimulus onset (**Figure 3D**) revealed a clear engagement of CaMKII-positive neurons from both hippocampal subregions during light–tone associations, which is constant across preconditioning sessions (**Figure 3E**).

## CaMKII-positive neurons in dorsal hippocampus mediate $_{LT}$SPC

The in vivo photometry results suggest a prominent engagement of the dorsal hippocampal (dHPC) subregion, characterized by increased activity of various hippocampal cell types, including CaMKII-positive neurons, during light–tone presentations in the preconditioning phase. However, no study to date has explored a potential causal dissociation between the roles of the dHPC and vHPC in mouse

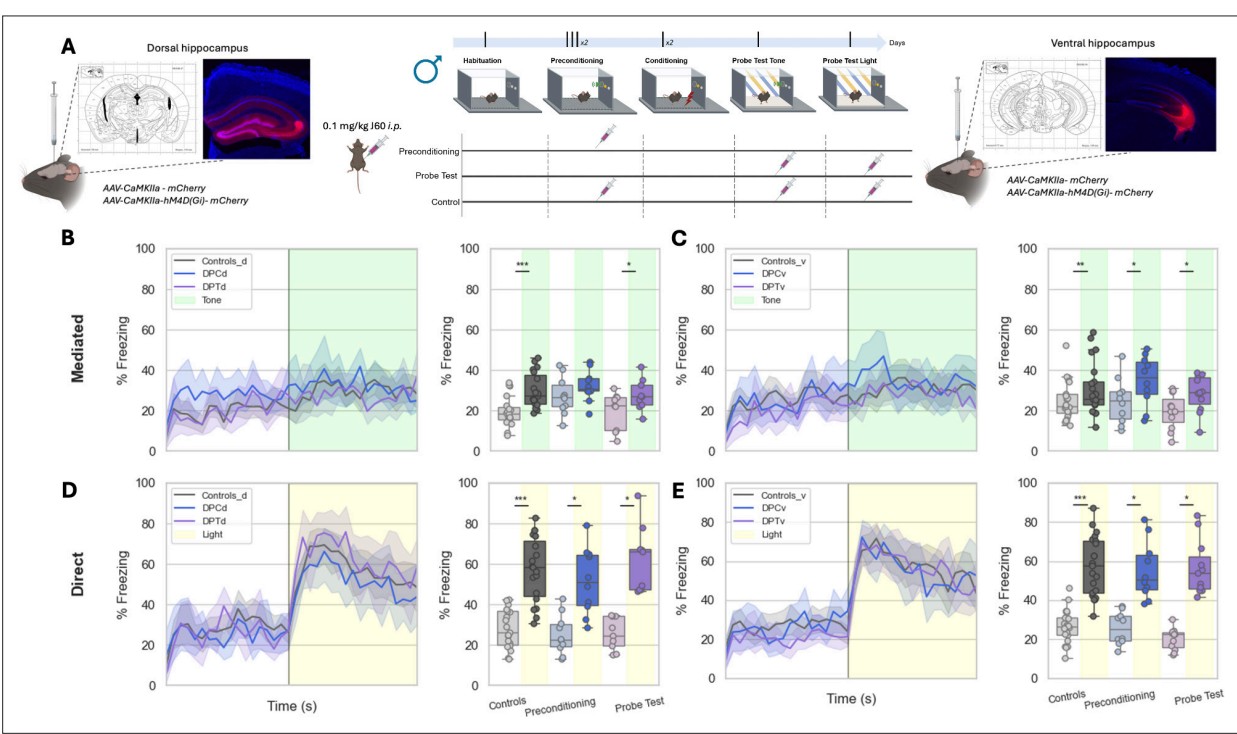

**Figure 4.** Chemogenetic modulation of dorsal and ventral hippocampus during $_{LT}$SPC. (**A**) Schematic representation of $_{LT}$SPC task combined with chemogenetic approaches, where an inhibitory DREADD was infused in dHPC (representative image on the left) and vHPC (representative image on the right). During the preconditioning (DPC) or the Probe Test (DPT), a DREADD agonist (J60) was injected intraperitoneally (controls were injected with the agonist in both phases) (diagram of DREADD agonist administration on the center). The temporal dynamics of freezing represented in bins of 10 s across time of experiment of the Probe Test 1 (tone) of dHPC (**B** on the left) and vHPC (**C** on the left) and Probe Test 2 (light) of dHPC (**D** on the left) and vHPC (**E** on the left). The percentage of time spent in freezing during OFF and ON periods (left) of the Probe Test 1 (tone) of animals infused in dHPC (**B** on the right) and vHPC (**C** on the right). The percentage of time spent in freezing during OFF and ON periods (left) of the Probe Test 2 (light) of animals infused in dHPC (**D** on the right) and vHPC (**E** on the right). Controls are labeleld as Controls_d (on dHPC) and Controls_V (on vHPC). The preconditioning groups are labeled as DPCd (on dHPC) and DPCv (on vHPC). The Probe Test groups are labeled as DPTd (on dHPC) and DPTv (on vHPC). *Significant p-value (<0.05) after false discovery rate (FDR). GLM: generalized linear model fitted to gamma distribution with planned comparisons. ** p<0,01; *** p<0,001. See statistical details in *Supplementary file 1A*.

The online version of this article includes the following figure supplement(s) for figure 4:

**Figure supplement 1.** Additional controls for chemogenetics experiments.

**Figure supplement 2.** Inhibition of PV interneurons in dorsal hippocampus during light–tone associations did not impact mediated learning.

**Figure supplement 3.** Temporal dynamics of the conditioning phase of all sensory preconditioning experiments.

sensory preconditioning. To investigate this, adeno-associated viral vectors expressing an inhibitory DREADD (hM4DGi, hereafter referred to as DREADD-Gi) (*Robinson and Adelman, 2015*) under the CaMKII promoter were infused into either the dHPC or vHPC to selectively inhibit principal excitatory neurons in these hippocampal subregions (*Figure 4A*). Control animals, which received either saline or the DREADD agonist J60, were pooled as they exhibited no significant behavioral differences and showed a reliable mediated and direct learning in the $_{LT}$SPC task (*Figure 4B* and, *Figure 4—figure supplement 1A–D*).

Notably, inhibition of CaMKII-positive neurons in the dHPC, which decrease calcium activity in CaMKII-positive neurons (*Figure 4—figure supplement 1E, F*), during the preconditioning phase (i.e., J60 administration in DREADD-Gi mice) completely abolished mediated learning (*Figure 4B*) without affecting direct learning (*Figure 4D*). This effect was specific to the preconditioning phase, as inhibition prior to conditioning sessions (*Figure 4—figure supplement 1G*) or before Probe Test 1 (*Figure 4B*) did not impair mediated learning. In contrast, inhibition of CaMKII-positive neurons in the vHPC, whether during preconditioning or Probe Test 1, had no effect on either mediated (*Figure 4C*) or direct learning (*Figure 4E*). These findings indicate that the dHPC, but not the vHPC, plays a specific and causal role in encoding associations between innocuous stimuli (e.g., light and tone) that underlie sensory preconditioning responses. Given the additional results observed in our photometry experiments and the crucial role of dHPC, we next investigated whether PV-positive interneurons, which were also activated during stimuli presentation in our photometry recordings, played a role in this behavior. To test this, a Cre-dependent DREADD-Gi virus was infused into the dHPC of PV-Cre mice to allow for selective inhibition of PV interneurons (*Figure 4—figure supplement 2A–C*) during light–tone pairings. Remarkably, silencing PV interneurons during the preconditioning phase did not affect either mediated or direct learning (*Figure 4—figure supplement 2D*).

Taken together, these results suggest that CaMKII-positive principal neurons, but not PV interneurons, in the dHPC are critical for encoding light–tone associations that drive sensory preconditioning responses in mice. Importantly, no statistical behavioral differences were observed during the conditioning across the different experiments (*Figure 4—figure supplement 3*).

## Discussion

Our study provides novel insights into the role of the hippocampus in higher-order conditioning by establishing and validating a $_{LT}$SPC task in male mice. Using in vivo calcium imaging, we linked the activity of specific hippocampal cells in the dorsal and ventral subregions to distinct behavioral phases of the task. Furthermore, chemogenetic inhibition experiments allowed us to causally identify the dorsal hippocampus (dHPC)—but not the ventral hippocampus (vHPC)—as essential for encoding light–tone pairings, which are critical for the behavioral outcomes observed in sensory preconditioning.

Sex differences in classical fear conditioning have been reported, with conflicting findings showing stronger conditioning in males, in females, or no differences at all (*Moore et al., 2010*; *Keiser et al., 2017*; *Anagnostaras et al., 1998*; *Maren et al., 1994*; *Colon et al., 2018*; *Urien et al., 2021*; *Kosten et al., 2006*; *Dachtler et al., 2011*). However, to our knowledge, no previous studies have examined sex differences in sensory preconditioning paradigms. Our results demonstrate that, using the same preconditioning and conditioning settings, only male mice show a reliable light–tone sensory preconditioning responding, whereas female mice show behavioral responses mostly associated with fear generalization. Indeed, our results could suggest that female mice are more vulnerable to aversive stimuli responding, which might be consistent with prior evidence showing sex-dependent differences in conditioned freezing responses (*Colom-Lapetina et al., 2019*; *Marcinkiewcz et al., 2019*; *Shansky and Murphy, 2021*; *Gruene et al., 2015*). When the light and the tone were presented separately during the preconditioning, or when the footshock was omitted, both mediated and direct learning were abolished in male mice. Instead, female mice still showed reduced but detectable cue-induced responses under these conditions, even if we increased the time between light and tone exposures during the preconditioning phase. Thus, we did not successfully set up a light–tone sensory preconditioning paradigm for female mice, and future research will be required to optimize and adapt a protocol to reliably observe mediated learning in females. Given that female mice also tend to exhibit greater fear generalization (*Keiser et al., 2017*; *Day et al., 2016*; *Krueger and Sangha, 2021*), the observed differences in the Unpaired group could reflect this enhanced generalization (*Urien et al., 2021*). In this regard, changes in the number of conditioning sessions and/or trials, intensity of the

electric footshock, or additional contextual changes could avoid this fear generalization in future sensory preconditioning paradigms in female mice. In addition, an interesting unresolved issue is how sex hormones could alter sensory preconditioning responding or the different performance of male and female mice in different protocols. Overall, our data clearly revealed the necessity to adapt behavioral protocols for each sex in sensory preconditioning paradigms involving auditory and visual stimuli and aversive conditioning.

Prior research has established the hippocampus as critical for encoding spatial location, context, cues, and memory related to appetitive and aversive behaviors (*Moser et al., 1995*; *Trouche et al., 2016*; *Trouche et al., 2019*; *Dragoi et al., 2003*; *Robinson et al., 2020*). However, few studies have addressed its causal involvement in reinforcement learning (*Ibrahim et al., 2024*), and even fewer have explored its role in forming associations between innocuous stimuli. Using in vivo fiber photometry, we observed increased neuronal activity in both the dorsal and ventral hippocampus during light–tone pairings and during light–shock associations. These findings support the notion that the hippocampus is actively engaged in encoding both conditioned (CS) and unconditioned (US) stimuli, in agreement with studies showing hippocampal activation during reinforcement behavior (*Ibrahim et al., 2024*; *Ursin et al., 1966*; *van der Kooy et al., 1977*). Still, some prior work has argued that the hippocampus is not required for forming simple cue–US associations, which may instead rely on the amygdala (*Kim and Fanselow, 1992*).

Our photometry results show that the simultaneous presentation of sensory cues is encoded by the hippocampus, reinforcing its role in associative learning and sensory integration (*Voss and Cohen, 2017*). This is consistent with previous work in both animals and humans supporting hippocampal involvement during associative phases of sensory preconditioning (*Wimmer and Shohamy, 2012*; *Busquets-Garcia et al., 2018*; *Iordanova et al., 2011*; *Wheeler et al., 2013*; *Talk et al., 2002*). Importantly, however, none of these earlier studies investigated the specific roles of hippocampal subregions or cell types. Our data indicate that modulation of hippocampal activity—specifically in the dHPC—affects encoding during the preconditioning phase but not during conditioning or retrieval phases. Chemogenetic inhibition of CaMKII-positive neurons in the dHPC during the preconditioning blocked mediated learning responding, while inhibition in the vHPC had no such effect. Moreover, silencing PV-positive interneurons in the dHPC during preconditioning did not disrupt sensory preconditioning, suggesting that principal excitatory neurons, but not PV interneurons, are essential for forming associations between innocuous stimuli. Indeed, it is somehow contradictory to observe an activity enhancement of PV-positive interneurons during light–tone associations and the lack of effect of its inhibition during this phase. To fully discard any role of the activity of these interneurons during this phase, future experiments should aim to particularly inhibit these cells during restricted temporal windows such as some preconditioning sessions rather than the full preconditioning phase or to specifically inhibit its activity during the light–tone exposures (e.g., closed-loop optogenetic approaches).

Other brain regions, including the orbitofrontal, retrosplenial, and perirhinal cortices, have also been implicated in processing associations between neutral sensory cues (*Holmes et al., 2013*; *Parkes and Westbrook, 2010*; *Robinson et al., 2014*; *Hart et al., 2020*). These areas maintain reciprocal anatomical connections with the hippocampus (*Agster and Burwell, 2013*; *Witter et al., 2017*; *Ritchey et al., 2015*), suggesting a broader, hippocampus-guided network for encoding higher-order associations. Our findings show that the dHPC—but not the vHPC—is crucial for encoding light–tone pairings may offer new insight into this circuit. A likely candidate for mediating such interactions is the retrosplenial cortex, which receives direct projections from the dHPC (*Opalka and Wang, 2020*). Thus, inhibiting CaMKII-positive neurons in the dHPC, which project to the retrosplenial cortex, could disrupt sensory preconditioning, supporting this potential circuit model. Future studies will be necessary to explore this hypothesis in detail.

Despite the robust findings presented by our data, several limitations must be acknowledged. First, while we successfully validated the $_{LT}$SPC protocol in male mice, further optimization is needed for its reliable use in female mice, as commented above. Second, the fixed order of Probe Tests may have introduced extinction effects for the tone (*Polack et al., 2013*), potentially affecting responses to the light cue. Future studies should counterbalance Probe Tests to address this possibility. Third, our behavioral protocol required context changes between experimental phases, which may have engaged hippocampal circuits (*Keinath et al., 2022*) more strongly than if a single context had been used. Thus, the impact of context-switching on hippocampal involvement should be addressed in

future work. Finally, this work has been focused on the activity of CaMKII-positive neurons and PV interneurons, although we cannot discard the role of other hippocampal cell types such as somatostatin interneurons, which have been recently involved in learning and associative processes (*Abbas et al., 2018*; *Lacagnina et al., 2024*). Future experiments will be required to fully characterize the role of different hippocampal subtypes in sensory preconditioning. Indeed, other techniques such as the optogenetic modulation of cell activity could give more details on the temporal engagement of these hippocampal circuits in higher-order conditioning.

In summary, our results identify a key role for hippocampal circuits—particularly CaMKII-positive neurons in the dorsal hippocampus—in mediating associations between innocuous stimuli during light–tone sensory preconditioning. This process reflects a fundamental cognitive mechanism underpinning higher-order learning and decision-making in both animals and humans. Understanding the neural substrates of sensory preconditioning is crucial, as this form of learning may contribute to adaptive behavior but also underlie maladaptive processes seen in disorders such as psychosis (*Busquets-Garcia et al., 2017b*; *Busquets-Garcia et al., 2017a*).

## Materials and methods

### Animals

Male and female C57BL/6J mice were purchased from Charles River Laboratories and were used for behavioral and chemogenetic experiments. Male Parvalbumin (PV)-Cre mice (The Jackson Laboratory, #017320), originally created in the laboratory of Dr. Silvia Arber (Friedrich Miescher Institute), were obtained by own breedings in our animal facility. All the mice used in this study were 8 weeks old at the beginning of the experiments, and they were grouped-housed and maintained in a temperature (20–24°C) and humidity (40–70%) controlled condition under a 12-hr light/dark cycle and had ad libitum access to food and water. All behavioral studies were performed during the dark cycle (from 9 am to 5 pm) by trained researchers who were blind to the different experimental conditions.

All experimental procedures shown in this work have followed European guidelines (2010/62/EU) and were approved by the Committee on Animal Health and Care of Barcelona Biomedical Research Park (PRBB) (ref. ABG-19-0055) and from the Generalitat de Catalunya (ref. 10784). All experiments were performed in the animal facility of the PRBB, which has the full accreditation from the Association for Assessment and Accreditation of Laboratory Animal Care (AAALAC).

### Light–tone sensory preconditioning task

The task was performed using automated conditioning chambers (Imetronic, France) and was divided into different phases (*Figure 1A*).

#### Habituation

Animals were exposed to the chambers with the presence of background noise (65 db) (like white noise but allocated to different speakers to avoid that animals associate the sound with a location) during one session of 20 min.

#### Preconditioning phase

This phase was composed of six sessions (2 per day; 3 hr of intersession interval) of 510 s each under background noise. These sessions started with an OFF period (only background noise) of 3 min and were followed by five simultaneous presentations of a light (CS1, white LED light of 1.8 cm of diameter located in one side of the box) and a tone (CS2, 65 db, 3000 Hz) during 30 s with an intertrial interval of 30 s and a final OFF period of 1 min. This protocol detail applies to the Paired and No-Shock experimental groups. For the unpaired groups, light and tone were never associated together during this phase. Indeed, animals were exposed to the same amount of time to light and tone but in different sessions (S1 light, S2 tone, S3 tone, S4 light, S5 light, S6 tone, in a pseudo-randomized way) separated by 3 hr (male and female mice) or 6 hr (female mice).

## Conditioning phase

This phase consisted of two training sessions in males and one training session in females. Each session lasted 510 s under background noise. In the case of males, there was an intersession interval of 3 hr. Specifically, the sessions started with an off period of 3 min and were followed by 5 presentations of a 10-s light stimulus (CS1) that co-terminated during 2 s with a mild footshock (0.4 mA, see results) with an intertrial interval of 1 min and a final OFF period of 1 min. This protocol detail applies to the Paired and Unpaired experimental groups. For the No-Shock group, animals followed the same conditioning session but without the exposure to the electric footshock. For the control group performed in female mice, no light was associated with the footshock during this conditioning session.

## Probe tests phase

Mice were subjected on the same day to two Probe Test sessions where they were exposed to the tone (mediated learning, Probe Test 1) and the light (direct learning, Probe Test 2) in a new context (different from the previous phases). We changed four features of the context: floor texture, wall design, smell (from 70% ethanol to a CR36 disinfectant solution), and absence of background noise to avoid fear responses elicited by the context itself. The two Probe Tests lasted 6 min and were separated by at least 1 hr. In this session, animals underwent an OFF period of 3 min followed by an ON period of 3 min where the tone (Probe Test 1) or the light (Probe Test 2) was continuously exposed. All experimental groups (Paired, Unpaired, and No-Shock) perform the Probe Tests in the same manner. The presence of mediated and direct learning is shown by the percentage of freezing during ON and OFF periods.

The behavior analysis was automatized using Deeplabcut to track the animal's position across time and a Python homemade script to compute all the different analyses, graphs, and statistics performed in the present paper. The main behavior measured was the freezing response (i.e., immobility), which was defined as an Euclidean distance lower than 0.02 cm per pair of frames for videos with 25 fps, resulting in a speed lower than 0.5 cm/s. We validated this automated behavioral counting by performing correlations with manual counts and another software dedicated to freezing scoring (EzTrack; *Pennington et al., 2021*), with which we found a high correlation (higher than 90%) (*Figure 1—figure supplement 1*). The Deeplabcut project and all the scripts are publically available in the GitHub repository of the lab (https://github.com/abusquets85/Pinho-et-al.-eLife-Manuscript-2025; copy archived at *Busquets-Garcia Lab, 2025*) including all the code creates to analyze behavior and photometry data.

## In vivo fiber photometry

Male C57BL/6J and PV-Cre mice were anesthetized with a mixture of ketamine (75 mg/kg, Imalgene 500, Merial, Spain) and medetomidine (1 mg/kg, Domtor, Spain) by intraperitoneal (i.p.) injection. Then, animals were placed into a stereotaxic apparatus (World Precision Instruments, FL, USA) with a mouse adaptor and lateral ear bars. For viral intra-hippocampus delivery, AAV vectors were injected with a Hamilton syringe coupled with a nanofill attached to a pump (UMP3-1, World Precision Instruments, FL, USA). PV-Cre mice were injected with a mixture of 200 nl of AAV.Syn.NES.jRCaMP1a. WPRE.SV40 (titer: $1 \times 10^{13}$, addgene 100848-AAV9) and 200 nl of a Cre-dependent AAV-syn-FLEX-jGCaMP8f-WPRE (titer: $1 \times 10^{13}$, addgene 162379-AAV9) in order to monitor both the activity of the synapsin-positive cells in red and the PV interneurons in green. C57BL/6J mice were injected with 400 nl of AAV-CaMKII-GCaMP6m (titer: $1 \times 10^{12}$, Tebu-BIO 189SL-AAV9) to monitor activity of the CaMKII-positive neurons in green. To check DREADD functionality, 200 nl of the AAV-CaMKII-GCaMP6m and 200 nl of the inhibitory DREADD were infused in a similar manner. These viral vectors were infused (1 nl/s) directly into the dorsal (in one hemisphere) or ventral hippocampus (in the other hemisphere) with the following coordinates in mm: dorsal, AP ±1.5, ML ±2, DV −1.5 and ventral, AP ±3.5, ML ±3.3, DV −3.5, according to Paxinos and Franklin brain atlas (*Paxinos and Franklin, 2001*). After the AAV infusions, an optic fiber (core 400 μm, N.A 0.5, RWD, China) was implanted using dental cement following the same coordinates except for DV that was placed 0.25 mm above viral infusions. Four weeks after this surgery, animals were used for in vivo calcium recordings where they underwent an $_{LT}$SPC, and in vivo recordings were performed during light–tone associations (preconditioning phase) and light–shock associations (conditioning phase). Before this experiment, animals were habituated for 3 days to connect and disconnect the optic fibers and to be habituated to the cable in the same

chamber where the behavioral experiment was performed. During these two behavioral phases, in vivo recordings were performed using a commercial fiber photometry equipment (RWD, China) where we used 470 nM LED to excite the GCaMP sensor, and 560 nM for the RCaMP signal. In all mice used, after the recording experiments, the signal was checked to validate the viral infusions.

To analyze the fiber photometry experiments, a custom Python code was used and the behavioral videos and photometry recordings were synchronized by TTL signals. Raw calcium signals were pre-processed by removing the first minute of the recording to decrease the effect of the initial exponential photobleaching and by removing point artifacts. The 470 nM signal was fitted to the isosbestic 405 nM using a linear fit, and for each time point, $\Delta F/F$ was calculated as ($F$470 nm − $F$405 nm (fitted))/$F$405 nm (fitted). This procedure is the same as described in previous works (*Lerner et al., 2015*). The codes used for the analysis will be found on the dedicated GitHub repository.

## Chemogenetic modulation

Stereotaxic surgeries were performed as described above for fiber photometry. In this case, C57BL/6J or PV-Cre mice were injected with 500 nl AAV-CaMKII-mCherry (titer: $7 \times 10^{12}$, 114469-AAV5) and 500 nl AAV-CaMKII-hM4Di (titer: $7 \times 10^{12}$, 50477-AAV2) directly into the hippocampus, with the following coordinates: dorsal, AP ±1.5, ML ±2, DV −1.5 and ventral, AP ±3.5, ML ±3.3, DV −3.5, according to Paxinos and Franklin brain atlas (*Paxinos and Franklin, 2001*). On the other hand, PV-Cre mice were infused with AAV-DIO-hM4Di using the same coordinates to target the dorsal hippocampus. Three control groups were performed for each subregion (dorsal and ventral hippocampus): animals infused with control virus (pAAV-CaMKIIa-mCherry) and injected with saline or JHU37160 dihydrochloride (J60, 0.1 mg/kg, i.p., HelloBio, HB6261) (*Bonaventura et al., 2019*), and animals infused with AAV-CaMKII-mCherry or AAV-DIO-hM4Di injected with saline. Animals were used for chemogenetics experiments 4 weeks after injections to get an optimal expression of the viruses. The injection of J60 was 1 hr before each preconditioning session (C57BL/6J and PV-Cre mice), before the Probe Test 1 (C57BL/6J mice) or the two conditioning sessions. In all mice used in the behavioral experiments, the signal was checked, and representative images are shown in Figure Supplements.

## Histology

After the chemogenetic and imaging experiments, mice were anesthetized i.p. with a mixture of ketamine (50 mg/kg) and xylazine (20 mg/kg) in overdose (3–4x body weight), transcardially perfused with cold 4% paraformaldehyde to fix tissues. Brains were removed and sectioned in serial coronal sections of 20 μm and collected directly to the slide for further analysis in the microscope. All sections were counterstained with 4',6-diamidino-2-phenylindole (DAPI, ref 00-4959-52, Fluoromount-G w/ DAPI, Life Technologies, USA) to visualize cell nuclei in the mounting medium. Slides were cover-slipped and imaged by an epifluorescence Nikon Eclipse Ni-E. All animals were signal checked to guarantee the target of the hippocampus subregion.

## Data collection and statistical analysis

### Data collection

All mice were randomly assigned to experimental conditions. Researchers performing the experiments were always blind to the condition of the subject, and we used an automated way to analyze behavioral responses throughout the study. Raw data was processed and analyzed using homemade Python packages. Each behavioral experiment was repeated two to three times in different animal batches, and the sample size was defined based on previous experiments by the authors.

### Statistical analysis

Graphs and statistical analysis were performed through homemade Python scripts that will be openly shared. All data comes from distinct mice and is shown as independent data points per animal ± SEM. Normality and homoscedasticity of the data were assessed with the Kolmogorov–Smirnov and Levene tests, respectively. Due to non-parametric properties of the data, a generalized linear model with gamma distribution was used for multivariable analysis (after a goodness of fit to select the appropriate distribution), followed by planned comparisons corrected with false discovery rate. For univariate analysis, the Kruskal–Wallis test was performed, followed by planned comparisons corrected with false discovery rate. For simple comparisons, the Mann–Whitney test was performed. Detailed

statistical data for each experiment can be found in *Supplementary file 1A* (for main figures) and *Supplementary file 1B* (for figure Supplements).

## Acknowledgements

We would like to thank the personnel of the Animal Facility of the Parc de Recerca Biomedica de Barcelona (PRBB) for mouse care. We thank all the members of our lab for useful discussions during the development of the project and Remi Proville (Aquineuro, Bordeaux, France) for the great help in the analysis of behavior and in vivo photometry. Finally, we would like to also thank Dr. Maria Victoria Puig Velasco for providing the PV-Cre colony. This work was supported by la Generalitat de Catalunya (SGR [00022] and 'Jo Investigo' [2022 INV-1 00005/100005TG3] programmes) from the Departament d'Economia i Coneixement de la Generalitat de Catalunya (Spain) and from the European Research Council (ERC) under the European Union's Horizon 2020 research and innovation programme (Grant Agreement No. 948217).

## Additional information

### Funding

| Funder | Grant reference number | Author |
| --- | --- | --- |
| Generalitat de Catalunya | SGR 00022 | Arnau Busquets-Garcia |
| Generalitat de Catalunya | Jo Investigo Program 2022 INV-1 00005/100005TG3 | Irene Manzanares-Sierra |
| European Research Council | 948217 | Arnau Busquets-Garcia |

The funders had no role in study design, data collection, and interpretation, or the decision to submit the work for publication.

### Author contributions

Julia S Pinho, Conceptualization, Resources, Data curation, Software, Formal analysis, Investigation, Visualization, Methodology, Writing – original draft, Writing – review and editing; Carla Ramon-Duaso, Irene Manzanares-Sierra, Data curation, Formal analysis, Investigation, Visualization, Methodology, Writing – original draft, Writing – review and editing; Arnau Busquets-Garcia, Conceptualization, Data curation, Formal analysis, Supervision, Funding acquisition, Investigation, Methodology, Writing – original draft, Project administration, Writing – review and editing

### Author ORCIDs

Julia S Pinho ⓘ https://orcid.org/0000-0002-8697-7888
Carla Ramon-Duaso ⓘ https://orcid.org/0000-0001-8358-4440
Irene Manzanares-Sierra ⓘ http://orcid.org/0009-0006-2461-5167
Arnau Busquets-Garcia ⓘ https://orcid.org/0000-0002-7100-8873

### Ethics

All experimental procedures shown in this work have followed European guidelines (2010/62/EU) and were approved by the Committee on Animal Health and Care of Barcelona Biomedical Research Park (PRBB) (ref. ABG-19-0055) and from the Generalitat de Catalunya (ref. 10784). All experiments were performed in the animal facility of the PRBB, which has the full accreditation from the Association for Assessment and Accreditation of Laboratory Animal Care (AAALAC).

Reviewer #1 (Public review): https://doi.org/10.7554/eLife.105863.3.sa1
Reviewer #2 (Public review): https://doi.org/10.7554/eLife.105863.3.sa2
Reviewer #3 (Public review): https://doi.org/10.7554/eLife.105863.3.sa3
Author response https://doi.org/10.7554/eLife.105863.3.sa4

# Additional files

## Supplementary files

MDAR checklist

Supplementary file 1. Statistical tables for main figures and figure supplements.

Source data 1. Raw data for main figures.

Source data 2. Raw data for figure supplements.

## Data availability

Programming scripts generated to analyze our behavioral and photometry data are collected in the GitHub of the lab (https://github.com/abusquets85/Pinho-et-al.-eLife-Manuscript-2025; copy archived at *Busquets-Garcia Lab, 2025*).

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
