## [Editor Report · eLife Assessment]

Pinho et al use in vivo calcium imaging and chemogenetic approaches to examine the involvement of hippocampal sub-regions across the different stages of a sensory preconditioning task in mice. They find evidence for sensory preconditioning in male mice. They also find that, in these mice, CaMKII-positive neurons in the dorsal hippocampus encode the audio-visual association that forms in stage 1 of the task. The evidence in support of these findings is **convincing**. The **important** study will be of interest to researchers in the fields of learning and memory and/or hippocampus function.

---

## [Referee Report · Reviewer #1 (Public review)]

Summary:

The study by Pinho et al. presents a novel behavioral paradigm for investigating higher-order conditioning in mice. The authors developed a task that creates associations between light and tone sensory cues, driving mediated learning. They observed sex differences in task acquisition, with females demonstrating faster mediated learning compared to males. Using fiber photometry and chemogenetic tools, the study reveals that the dorsal hippocampus (dHPC) plays a central role in encoding mediated learning. These findings are crucial for understanding how environmental cues, which are not directly linked to positive/negative outcomes, contribute to associative learning. Overall, the study is well-designed, with robust results, and the experimental approach aligns with the study's objectives.

Strengths:

The authors develop a robust behavioral paradigm to examine higher-order associative learning in mice.

They discover a sex-specific component influencing mediated learning.

Using fiber photometry and chemogenetic techniques, the authors identify the dorsal hippocampus but not the ventral hippocampus, plays a crucial for encoding mediated learning.

---

## [Referee Report · Reviewer #2 (Public review)]

Pinho et al. developed a new auditory-visual sensory preconditioning procedure in mice. They observed sex differences in this task, with male, but not female mice acquiring preconditioned fear. Using photometry, they observed activation of the dorsal and ventral hippocampus during sensory preconditioning (tone + light) and direct conditioning (light + shock). Finally, the authors combined their sensory preconditioning task with DREADDs. They found that inhibition of CamKII-positive cells in the dorsal hippocampus, but not the ventral hippocampus, during the preconditioning phase impaired the formation of sensory preconditioned fear. However, inhibiting the same cells during phase two (light + shock) had no effect.

Strengths:

(1) The authors develop a robust auditory-visual sensory preconditioning protocol in male mice. Research on the neurobiology of sensory preconditioning has primarily used rats as subjects. The development of a mouse protocol will be very beneficial to the field, allowing researchers to take advantage of the many transgenic mouse lines.

(2) They find sex differences in the acquisition of sensory preconditioning, raising the importance of adapting behavioral procedures to sex

(3) They identify the dorsal (but not ventral) hippocampus as a key region for the integration of sensory information during the preconditioning phase, furthering our understanding of the role of the hippocampus in integrating experience.

Comments on the revisions:

Thank you for addressing my concerns in considerable detail. I have no more suggestions for the authors.

---

## [Referee Report · Reviewer #3 (Public review)]

Summary:

Pinho et al., investigated the role of the dorsal VS ventral hippocampus and gender differences in mediated learning. While previous studies already established the engagement of the hippocampus in sensory preconditioning, the authors here took advantages of freely-moving fiber photometry recording and chemogenetics to observe and manipulate sub-regions of the hippocampus (drosal VS ventral) in a cell-specific manner. Importantly, the authors validated the sensory preconditioning procedure in male mice. The authors found no evidence of sensory preconditioning in female mice, but rather a generalization effect, stressing the importance of gender differences in fear learning. After validation of a sensory preconditioning procedure in male mice using light and tone neutral stimuli and a mild foot shock as the unconditioned stimulus, the authors used fiber photometry to record from all neurons VS parvalbumin_positive_only neurons in the dorsal hippocampus or ventral hippocampus of male mice during both preconditioning and conditioning phases. They found an increased activity of all neurons, PV+_only neurons, and CAMKII+ neurons in both sub-regions of the hippocampus during both preconditioning and conditioning phases. Finally, the authors found that chemogenetic inhibition of CaMKII+ neurons (but not PV+_only neurons) in the dorsal (but not ventral) hippocampus specifically prevented the formation of an association between the two neutral stimuli (i.e., light and tone cues). This manipulation had no effect on the direct association between the light cue and the mild foot shock. This set of data (1) validates sensory preconditioning in male mice, and stresses the importance of taking gender effect into account; (2) validates the recruitment of dorsal and ventral hippocampi during preconditioning and conditioning phases; (3) and further establishes the specific role of CaMKII+ neurons in the dorsal hippocampus, but not ventral hippocampus, in the formation of an association between two neutral stimuli, but not between a neutral-stimulus and a mild foot shock.

Strengths:

The authors developed a sensory preconditioning procedure in male mice to investigate mediated learning using light and tone cues as neutral stimuli, and a mild foot shock as the unconditioned stimulus. They provide evidence of a gender effect in the formation of light-cue association. The authors took advantage of fiber-photometry and chemogenetics to target sub-regions of the hippocampus, in a cell-specific manner and investigate their role during different phases of a sensory conditioning procedure, and developed a DeepLabCut-based strategy to assess freezing fear responses.

Weaknesses:

The authors went further than previous studies by investigating the role of sub-regions the hippocampus in mediated learning, however, there are a few weaknesses that should be addressed in future studies:

(1) This study found a generalization effect in female mice only. While the authors attempted to neutralize this effect, the mechanism underlying this gender effect and whether female mice can display evidence for mediated learning has yet to be determined.

(2) One of the main effects from which derives the conclusion of this study (i.e., deficit of mediated learning in male mice when CAMKII+ neurons are inhibited in the dorsal HPC during the preconditioning phase) lies in the absence of a significant difference of the freezing response before and during the tone cue presentation when CAMKII+ are chemogenetically inhibited during the Probe Test Tone phase (cf. Fig. 4 Panel B, DPCd group). The fear response before the tone cue presentation in this group (DPCd) seems higher than in Controls_d and DPTd groups and could have masked a mediated learning effect.

---

## [Author Response]

The following is the authors’ response to the original reviews.

**Reviewer #1 (Public review):**
Summary:The study by Pinho et al. presents a novel behavioral paradigm for investigating higher-order conditioning in mice. The authors developed a task that creates associations between light and tone sensory cues, driving mediated learning. They observed sex differences in task acquisition, with females demonstrating faster-mediated learning compared to males. Using fiber photometry and chemogenetic tools, the study reveals that the dorsal hippocampus (dHPC) plays a central role in encoding mediated learning. These findings are crucial for understanding how environmental cues, which are not directly linked to positive/negative outcomes, contribute to associative learning. Overall, the study is well-designed, with robust results, and the experimental approach aligns with the study's objectives.Strengths:(1) The authors develop a robust behavioral paradigm to examine higher-order associative learning in mice.(2) They discover a sex-specific component influencing mediated learning, with females exhibiting enhanced learning abilities.(3) Using fiber photometry and chemogenetic techniques, the authors identify the dorsal hippocampus but not the ventral hippocampus, which plays a crucial for encoding mediated learning.

We appreciate the strengths highlighted by the Reviewer and the valuable and complete summary of our work.

Weaknesses:(1) The study would be strengthened by further elaboration on the rationale for investigating specific cell types within the hippocampus.

We thank the Reviewer for highlighting this important point. In the revised manuscript, we have added new information (Page 11, Lines 27-34) to specifically explain the rational of studying the possible cell-type specific involvement in sensory preconditioning.

(2) The analysis of photometry data could be improved by distinguishing between early and late responses, as well as enhancing the overall presentation of the data.

According to the Reviewer comment, we have included new panels in Figure 3E and the whole Supplementary Figure 4, which separates the photometry data across different preconditioning and conditioning sessions, respectively. Overall, this data suggests that there are no major changes on cell activity in both hippocampal regions during the different sessions as similar light-tone-induced enhancement of activity is observed. These findings have been incorporated in the Results Section (Page 12, Lines 13-15, 19-20 and 35-36).

(3) The manuscript would benefit from revisions to improve clarity and readability.

Based on the fair comment, we have gone through the text to increase clarity and readability.

**Reviewer #2 (Public review):**
Summary:Pinho et al. developed a new auditory-visual sensory preconditioning procedure in mice and examined the contribution of the dorsal and ventral hippocampus to learning in this task. Using photometry they observed activation of the dorsal and ventral hippocampus during sensory preconditioning and conditioning. Finally, the authors combined their sensory preconditioning task with DREADDs to examine the effect of inhibiting specific cell populations (CaMKII and PV) in the DH on the formation and retrieval/expression of mediated learning.Strengths:The authors provide one of the first demonstrations of auditory-visual sensory preconditioning in male mice. Research on the neurobiology of sensory preconditioning has primarily used rats as subjects. The development of a robust protocol in mice will be beneficial to the field, allowing researchers to take advantage of the many transgenic mouse lines. Indeed, in this study, the authors take advantage of a PV-Cre mouse line to examine the role of hippocampal PV cells in sensory preconditioning.

We acknowledge the Reviewer´s effort and for highlighting the strengths of our work.

Weaknesses:(1) The authors report that sensory preconditioning was observed in both male and female mice. However, their data only supports sensory preconditioning in male mice. In female mice, both paired and unpaired presentations of the light and tone in stage 1 led to increased freezing to the tone at test. In this case, fear to the tone could be attributed to factors other than sensory preconditioning, for example, generalization of fear between the auditory and visual stimulus.

We thank the comment raised by the Reviewer. At first, we were hypothesizing that female mice were somehow able to associate light and tone although they were presented separately during the preconditioning sessions. Thus, we designed new experiments (shown in Supplementary Figure 2D) to test if we would observe data congruent with our initial hypothesis or with fear generalization as proposed by the reviewer. We have performed a new experiment comparing a Paired group with two additional control groups that are (i) an Unpaired group where we increased the time between the light and tone presentations and (ii) an experimental group where the light was absent during the conditioning. Clearly, the new results indicate the presence of fear generalization in female mice aswe found a significant cue-induced increase on freezing responses in all the experimental groups tested. In accordance with the Reviewer’s suggestion, we can conclude that mediated learning is not correctly observed in female mice using the protocol described (i.e. with 2 conditioning sessions). All these new results forced us to reorganize the structure and the figures of the manuscript to focus more in male mice in the Main Figures whereas showing the data with female mice in Supplementary Figures. Overall, our data clearly revealed the necessity to have adapted behavioral protocols for each sex demonstrating sex differences in sensory preconditioning, which was added in the Discussion Section (Page 15, lines 12-37).

(2) In the photometry experiment, the authors report an increase in neural activity in the hippocampus during both phase 1 (sensory preconditioning) and phase 2 (conditioning). In the subsequent experiment, they inhibit neural activity in the DH during phase 1 (sensory preconditioning) and the probe test, but do not include inhibition during phase 2 (conditioning). It was not clear why they didn't carry forward investigating the role of the hippocampus during phase 2 conditioning. Sensory preconditioning could occur due to the integration of the tone and shock during phase two, or retrieval and chaining of the tonelight-shock memories at test. These two possibilities cannot be differentiated based on the data. Given that we do not know at which stage the mediate learning is occurring, it would have been beneficial to additionally include inhibition of the DH during phase 2.

Following the Reviewer’s valuable comment, we have conducted a new experiment where we have chemogenetically inhibited the CaMKII-positive neurons of the dHPC during the conditioning to explore their involvement in mediated learning formation. Notably, the inhibition of principal neurons of the dHPC during conditioning does not impair the formation ofthe mediated learning in our hands. These new results are now shown in Supplementary Figure 7G and added in the Results section (Page 13, Lines 19-23).

(3) In the final experiment, the authors report that inhibition of the dorsal hippocampus during the sensory preconditioning phase blocked mediated learning. While this may be the case, the failure to observe sensory preconditioning at test appears to be due more to an increase in baseline freezing (during the stimulus off period), rather than a decrease in freezing to the conditioned stimulus. Given the small effect, this study would benefit from an experiment validating that administration of J60 inhibited DH cells. Further, given that the authors did not observe any effect of DREADD inhibition in PV cells, it would also be important to validate successful cellular silencing in this protocol.

According to the Reviewer comments, we have performed new experiments to validate the use of J60 to inhibit hippocampal cells that are shown in Supplementary Figure 7 E-F for CaMKII-positive neurons, in which J60 administration tends to decrease the frequency of calcium events both in the dHPC and vHPC. Furthermore, in Supplementary Figure 8 B-C we show that J60 is also able to modify calcium events in PV-positive interneurons. Although,the best method to validate the use of DREADD (i.e. to inhibit hippocampal cell activity) could be electrophysiology recordings, we lack this technique in our laboratory. Thus, in order to adress the reviewer comment, we decided to combine the DREADD modulation through J60 administration with photometry recordings, where several tendencies are confirmed. In addition, a similar approach has been used in another preprint of the lab (https://doi.org/10.1101/2025.08.29.673009), where there is an increase of phospho-PDH, a marker of neuronal inhibition upon J60 administration in the dHPC, as well as in other experiments conducted from a collaborator lab where they were able to observe a modulation of SOM-positive interneurons activity upon J60 administration (PhD defense of Miguel Sabariego, University Pompeu Fabra, Barcelona).

**Reviewer #3 (Public review):**
Summary:Pinho et al. investigated the role of the dorsal vs ventral hippocampus and the gender differences in mediated learning. While previous studies already established the engagement of the hippocampus in sensory preconditioning, the authors here took advantage of freely-moving fiber photometry recording and chemogenetics to observe and manipulate sub-regions of the hippocampus (dorsal vs. ventral) in a cell-specific manner. The authors first found sex differences in the preconditioning phase of a sensory preconditioning procedure, where males required more preconditioning training than females for mediating learning to manifest, and where females displayed evidence of mediated learning even when neutral stimuli were never presented together within the session.After validation of a sensory preconditioning procedure in mice using light and tone neutral stimuli and a mild foot shock as the unconditioned stimulus, the authors used fiber photometry to record from all neurons vs. parvalbumin_positive_only neurons in the dorsal hippocampus or ventral hippocampus of male mice during both preconditioning and conditioning phases. They found increased activity of all neurons, as well as PV+_only neurons in both sub-regions of the hippocampus during both preconditioning and conditioning phases. Finally, the authors found that chemogenetic inhibition of CaMKII+ neurons in the dorsal, but not ventral, hippocampus specifically prevented the formation of an association between the two neutral stimuli (i.e., light and tone cues), but not the direct association between the light cue and the mild foot shock. This set of data: (1) validates the mediated learning in mice using a sensory preconditioning protocol, and stresses the importance of taking sex effect into account; (2) validates the recruitment of dorsal and ventral hippocampi during preconditioning and conditioning phases; and (3) further establishes the specific role of CaMKII+ neurons in the dorsal but not ventral hippocampus in the formation of an association between two neutral stimuli, but not between a neutralstimulus and a mild foot shock.Strengths:The authors developed a sensory preconditioning procedure in mice to investigate mediated learning using light and tone cues as neutral stimuli, and a mild foot shock as the unconditioned stimulus. They provide evidence of a sex effect in the formation of light-cue association. The authors took advantage of fiber-photometry and chemogenetics to target sub-regions of the hippocampus, in a cell-specific manner and investigate their role during different phases of a sensory conditioning procedure.

We thank the Reviewer for the extensive summary of our work and for giving interesting value to some of our findings.

Weaknesses:The authors went further than previous studies by investigating the role of sub-regions of the hippocampus in mediated learning, however, there are several weaknesses that should be noted:(1) This work first validates mediated learning in a sensory preconditioning procedure using light and tone cues as neutral stimuli and a mild foot shock as the unconditioned stimulus, in both males and females. They found interesting sex differences at the behavioral level, but then only focused on male mice when recording and manipulating the hippocampus. The authors do not address sex differences at the neural level.

We appreciate the comment of the Reviewer. Indeed, thanks to other Reviewer comments during this revision process (see Point 1 of Reviewer #2), we performed an additional experiment that reveals that using the described protocol in female mice we observed fear generalization rather than mediated learning responding. This data pointed to the need of sex-specific changes in the behavioral protocols to measure sensory preconditioning. The revised version of the manuscript, although highlighting these sex differences in behavioral performance (see Supplementary Figure 2), is more focused in male mice and, accordingly, all photometry or chemogenetic experiments are performed using male mice. In future studies, once we are certain to have a sensory preconditioning paradigm working in female mice, it will be very interesting to study if the same hippocampal mechanisms mediating this behavior in male mice are also observed in female mice.

(2) As expected in fear conditioning, the range of inter-individual differences is quite high. Mice that didn't develop a strong lightshock association, as evidenced by a lower percentage of freezing during the Probe Test Light phase, should manifest a low percentage of freezing during the Probe Test Tone phase. It would interesting to test for a correlation between the level of freezing during mediated vs test phases.

Thanks to the comment raised by the reviewer, we generated a new set of data correlating mediated and direct fear responses. As it can be observed in Supplementary Figure 3, there is a significant correlation between mediated and direct learning in male mice (i.e. the individuals that freeze more in the direct learning test, correlate with the individuals that express more fear response in the mediated learning test). In contrast, this correlation is absent in female mice, further confirming what we have explained above. We have highlighted this new analysis in the Results section (Page 11, Lines 20-24).

(3) The use of a synapsin promoter to transfect neurons in a non-specific manner does not bring much information. The authors applied a more specific approach to target PV+ neurons only, and it would have been more informative to keep with this cell-specific approach, for example by looking also at somatostatin+ inter-neurons.

The idea behind using a pan neuronal promoter was to assess in general terms how neuronal activity in the hippocampus is engaged during different phases of the lighttone sensory preconditioning. However, the comment of the Reviewer is very pertinent and, as suggested, we have generated some new data targeting CaMKII-positive neurons (see Point 4 below). Finally, although it could be extremely interesting, we believe that targeting different interneuron subtypes is out of the scope of the present work. However, we have added this in the Discussion Section as a future perspective/limitation of our study (Page 17, Lines 9-24).

(4) The authors observed event-related Ca2+ transients on hippocampal pan-neurons and PV+ inter-neurons using fiber photometry. They then used chemogenetics to inhibit CaMKII+ hippocampal neurons, which does not logically follow. It does not undermine the main finding of CaMKII+ neurons of the dorsal, but not ventral, hippocampus being involved in the preconditioning, but not conditioning, phase. However, observing CaMKII+ neurons (using fiber photometry) in mice running the same task would be more informative, as it would indicate when these neurons are recruited during different phases of sensory preconditioning. Applying then optogenetics to cancel the observed event-related transients (e.g., during the presentation of light and tone cues, or during the foot shock presentation) would be more appropriate.

We have generated new photometry data to analyze the activity of CaMKII-positive neurons during the preconditioning phase to confirm their engagement during the light-tone pairings. Thus, we infused a CaMKII-GCAMP calcium sensor into the dHPC and vHPC of mice and we recorded its activity during the 6 preconditioning sessions. The new results can be found in Figure 3 and explained in the Results section (Page 12, Lines 26-36). The results clearly show an engagement of CaMKII-positive neurons during the light-tone pairing observed both in the dHPC and vHPC. Finally, although the suggestion of performing optogenetic manipulations would be very elegant, we expect to have convinced the reviewer that our chemogenetic results clearly show and are enough to demonstrate the involvement of dHPC in the formation of mediated learning in the Light-Tone sensory preconditioning paradigm. However, we have added this in the Discussion Section as a future perspective/limitation of our study (Page 17, Lines 9-24).

(5) Probe tests always start with the "Probe Test Tone", followed by the "Probe Test Light". "Probe Test Tone" consists of an extinction session, which could affect the freezing response during "Probe Test Light" (e.g., Polack et al. (http://dx.doi.org/10.3758/s13420-013-0119-5)). Preferably, adding a group of mice with a Probe Test Light with no Probe Test Tone could help clarify this potential issue. The authors should at least discuss the possibility that the tone extinction session prior to the "Probe Test Light" could have affected the freezing response to the light cue.

We appreciate the comment raised by the reviewer. However, we think that our direct learning responses are quite robust in all of our experiments and, thus, the impact of a possible extinction based on the tone presentation should not affect our direct learning. However, as it is an important point, we have discussed it in the Discussion Section (Page 17, Lines 12-14).

**Reviewer #4 (Public review):**
SummaryPinho et al use in vivo calcium imaging and chemogenetic approaches to examine the involvement of hippocampal sub-regions across the different stages of a sensory preconditioning task in mice. They find clear evidence for sensory preconditioning in male but not female mice. They also find that, in the male mice, CaMKII-positive neurons in the dorsal hippocampus: (1) encode the audio-visual association that forms in stage 1 of the task, and (2) retrieve/express sensory preconditioned fear to the auditory stimulus at test. These findings are supported by evidence that ranges from incomplete to convincing. They will be valuable to researchers in the field of learning and memory.

We appreciate the summary of our work and all the constructive comments raised by the Reviewer, which have greatly improved the clarity and quality of our manuscript.

AbstractPlease note that sensory preconditioning doesn't require the stage 1 stimuli to be presented repeatedly or simultaneously.

The reviewer is right, and we have corrected and changed that information in the revised abstract.

"Finally, we combined our sensory preconditioning task with chemogenetic approaches to assess the role of these two hippocampal subregions in mediated learning." This implies some form of inhibition of hippocampal neurons in stage 2 of the protocol, as this is the only stage of the protocol that permits one to make statements about mediated learning. However, it is clear from what follows that the authors interrogate the involvement of hippocampal sub-regions in stages 1 and 3 of the protocol - not stage 2. As such, most statements about mediated learning throughout the paper are potentially misleading (see below for a further elaboration of this point). If the authors persist in using the term mediated learning to describe the response to a sensory preconditioned stimulus, they should clarify what they mean by mediated learning at some point in the introduction. Alternatively, they might consider using a different phrase such as "sensory preconditioned responding".

Considering the arguments of the Reviewer, we have modified our text in the Abstract and through the main text. Moreover, based on a comment of Reviewer #2 (Point 2) we have generated new data demonstrating that dHPC does not seem to be involved in mediated learning formation during Stage 2, as its inhibition does not impair sensory preconditioning responding. This new data can be seen in Supplementary Figure 7G.

Introduction"Low-salience" is used to describe stimuli such as tone, light, or odour that do not typically elicit responses that are of interest to experimenters. However, a tone, light, or odour can be very salient even though they don't elicit these particular responses. As such, it would be worth redescribing the "low-salience" stimuli in some other terms.

Through the revised version of the manuscript, we have replaced the term “lowsalience” by “innocuous stimuli” or avoiding any adjective as we think is not necessary.

"These higher-order conditioning processes, also known as mediated learning, can be captured in laboratory settings through sensory preconditioning procedures2,6-11." Higher-order conditioning and mediated learning are not interchangeable terms: e.g., some forms of second-order conditioning are not due to mediated learning. More generally, the use of mediated learning is not necessary for the story that the authors develop in the paper and could be replaced for accuracy and clarity. E.g., "These higher-order conditioning processes can be studied in the laboratory using sensory preconditioning procedures2,6-11."

According to the Reviewer proposal, we have modified the text.

In reference to Experiment 2, it is stated that: "However, when light and tone were separated on time (Unpaired group), male mice were not able to exhibit mediated learning response (Figure 2B) whereas their response to the light (direct learning) was not affected (Figure 2D). On the other hand, female mice still present a lower but significant mediated learning response (Figure 2C) and normal direct learning (Figure 2E). Finally, in the No-Shock group, both male (Figure 2B and 2D) and female mice (Figure 2C and 2E) did not present either mediated or direct learning, which also confirmed that the exposure to the tone or light during Probe Tests do not elicit any behavioral change by themselves as the presence of the electric footshock is required to obtain a reliable mediated and direct learning responses." The absence of a difference between the paired and unpaired female mice should not be described as "significant mediated learning" in the latter. It should be taken to indicate that performance in the females is due to generalization between the tone and light. That is, there is no sensory preconditioning in the female mice. The description of performance in the No-shock group really shouldn't be in terms of mediated or direct learning: that is, this group is another control for assessing the presence of sensory preconditioning in the group of interest. As a control, there is no potential for them to exhibit sensory preconditioning, so their performance should not be described in a way that suggests this potential.

All these comments are very pertinent and also raised by Reviewer #2 (Point 1, see above). In the revised version of the manuscript, we have carefully changed, when necessary, our interpretation of the results (e.g. in the case of the No-Shock group). In addition, we have generated new data that confirm that using similar conditions (i.e. 2 conditioning sessions in our SPC) in female mice we observe fear generalization and not a confident sensory preconditioning responding. In our opinion, this is not discarding the presence of mediated learning in female mice but suggesting that adapted protocols must be used in each sex. These results forced us to change the organization of the Figures but we hope the reviewer would agree with all the changes proposed. In addition, we have re-wrote a paragraph in the Discussion Section to explain these sex differences (see Page 15, lines 12-37).

Methods - BehaviorI appreciate the reasons for testing the animals in a new context. This does, however, raise other issues that complicate the interpretation of any hippocampal engagement: e.g., exposure to a novel context may engage the hippocampus for exploration/encoding of its features - hence, it is engaged for retrieving/expressing sensory preconditioned fear to the tone. This should be noted somewhere in the paper given that one of its aims is to shed light on the broader functioning of the hippocampus in associative processes.This general issue - that the conditions of testing were such as to force engagement of the hippocampus - is amplified by two further features of testing with the tone. The first is the presence of background noise in the training context and its absence in the test context. The second is the fact that the tone was presented for 30 s in stage 1 and then continuously for 180s at test. Both changes could have contributed to the engagement of the hippocampus as they introduce the potential for discrimination between the tone that was trained and tested.

We have now added these pertinent comments in a “Study limitations” paragraph found in the Discussion Section (Page 17, Lines 9-24). Indeed, the different changes of context (including the presence of background noise) have been implemented by the fact that during the setting up of the paradigm we had problems of fear generalization (also in male mice). Similarly, differences in cue exposure between the preconditioning phase and the test phase were also decided based on important differences between previous protocols used in rats compared to how mice are responding. Certainly, mice were not able to adapt their behavioral responses when shorter time windows exposing the cue were used as it clearly happens with rats [1].

Results - BehaviorThe suggestion of sex differences based on differences in the parameters needed to generate sensory preconditioning is interesting. Perhaps it could be supported through some set of formal analyses. That is, the data in supplementary materials may well show that the parameters needed to generate sensory preconditioning in males and females are not the same. However, there needs to be some form of statistical comparison to support this point. As part of this comparison, it would be neat if the authors included body weight as a covariate to determine whether any interactions with sex are moderated by body weight.

Regarding the comparison between male and female mice, although the comments of the Reviewer are pertinent and interesting, we think that with the new data generated is not appropriate to compare both sexes as we still have to optimize the SPC protocol for female mice.

What is the value of the data shown in Figure 1 given that there are no controls for unpaired presentations of the sound and light? In the absence of these controls, the experiment cannot have shown that "Female and male mice show mediated learning using an auditory-visual sensory preconditioning task" as implied by its title. Minimally, this experiment should be relabelled.

Based on the new data generated with female mice, we have decided to remove Figure 1 and re-organize the structure of the manuscript. We hope that the Reviewer would agree that this has improved the clarity of the manuscript.

"Altogether, this data confirmed that we successfully set up an LTSPC protocol in mice and that this behavioral paradigm can be used to further study the brain circuits involved in higherorder conditioning." Please insert the qualifier that LTSPC was successfully established in male mice. There is no evidence of LTSPC in female mice.

We fully agree with the Reviewer and our new findings further confirm this issue. Thus, we have changed the statement in the revised version of the manuscript.

Results - Brain"Notably, the inhibition of CaMKII-positive neurons in the dHPC (i.e. J60 administration in DREADD-Gi mice) during preconditioning (Figure 4B), but not before the Probe Test 1 (Figure 4B), fully blocked mediated, but not direct learning (Figure 4D)." The right panel of Figure 4B indicates no difference between the controls and Group DPC in the percent change in freezing from OFF to ON periods of the tone. How does this fit with the claim that CaMKII-positive neurons in the dorsal hippocampus regulate associative formation during the session of tone-light exposures in stage 1 of sensory preconditioning?

To improve the quality of the figures and to avoid possible redundancies between panels, in the new version of the manuscript, we have decided to remove all the panels regarding the percentage of change. However, in our opinion regarding the issue raised by the Reviewer, the inhibition of the dHPC clearly induced an impairment of mediated learning as animals do not change their behavior (i.e. there is no significant increase of freezing between OFF and ON periods) when the tone appears in comparison with the other two groups. The graphs indicating the percentage of change (old version of the manuscript) was a different manner to show the presence of tone- or light-induced responses in each experimental group. Thus, a significant effect (shown by # symbol) meant that in that specific experimental group there was a significant change in behavior (freezing) when the cue (tone or light) appeared compared when there was no cue (OFF period). Thus, in the old panel 4B commented by the Reviewer, in our opinion, the absence of significance in the group where the dHPC has been inhibited during thepreconditioning, compared to the other groups, where a clear significant effect can be observed, indicate an impairment of mediated learning formation. However, to avoid any confusion, we have slightly modified the text to strictly mention what is being analyzed and/or shown in the graphs and, as mentioned, the graphs of percentage of change have been removed.

Discussion"When low salience stimuli were presented separated on time or when the electric footshock was absent, mediated and direct learning were abolished in male mice. In female mice, although light and tone were presented separately during the preconditioning phase, mediated learning was reduced but still present, which implies that female mice are still able to associate the two low-salience stimuli."

This doesn't quite follow from the results. The failure of the female unpaired mice to withhold their freezing to the tone should not be taken to indicate the formation of a light-tone association across the very long interval that was interpolated between these stimulus presentations. It could and should be taken to indicate that, in female mice, freezing conditioned to the light simply generalized to the tone (i.e., these mice could not discriminate well between the tone and light).

As discussed above, we fully agree with the Reviewer and all the manuscript has been modified as described above.

"Indeed, our data suggests that when hippocampal activity is modulated by the specific manipulation of hippocampal subregions, this brain region is not involved during retrieval." Does this relate to the results that are shown in the right panel of Figure 4B, where there is no significant difference between the different groups? If so, how does it fit with the results shown in the left panel of this figure, where differences between the groups are observed?"In line with this, the inhibition of CaMKII-positive neurons from the dorsal hippocampus, which has been shown to project to the restrosplenial cortex56, blocked the formation of mediated learning."Is this a reference to the findings shown in Figure 4B and, if so, which of the panels exactly? That is, one panel appears to support the claim made here while the other doesn't. In general, what should the reader make of data showing the percent change in freezing from stimulus OFF to stimulus ON periods?

In our opinion, as pointed above, the graphs indicating the percentage of change were a different manner to show the presence of tone- or light-induced behavioral responses in each experimental group. Thus, a significant effect (shown by # symbol) meant that in this specific experimental group there was a significant change in behavior (freezing) when the cue (tone or light appear) compared when there was no cue (OFF period). Thus, in the old panel 4B commented by the Reviewer, in our opinion, the absence of significance in the group where the dHPC has been inhibited during the preconditioning, compared to the other groups where a clear significant effect can be observed, indicates an impairment of mediated learning formation. In the revised version of the manuscript, we have rephrased these sentences to stick to what the graphs are showing and, as explained, the graphs of percentage of change have been removed.

**Reviewer #1 (Recommendations for the authors):**
The authors may address the following questions:(1) The study identifies major sex differences in the conditioning phase, with females showing faster learning. Since hormonal fluctuations can influence learning and behavior, it would be helpful for the authors to comment on whether they tracked the estrous cycle of the females and whether any potential effects of the cycle on mediated learning were considered.

This is a relevant and important point raised by the Reviewer. In our study we did not track the estrous cycle to investigate whether it exists any effect of the cycle on mediated learning, which could be an interesting project by itself. Although in the revised version of the manuscript we provide new information regarding the mediated learning performance in male and female mice, we agree with the reviewer that sex hormones may account for the observed sex differences. However, the aim of the present work was to explore potential sex differences in mediated learning responding rather than to investigate the specific mechanisms behind these potential sex differences.

For this reason and to avoid adding further complexity to our present study, we did not check the estrous cycle in the female mice, the testosterone levels in male mice or analyze the amount of sex hormones during different phases of the sensory preconditioning task. Indeed, we think that checking the estrous cycle in female mice would still not be enough to ascertain the role of sex hormones because checking the androgen levels in male mice would also be required. In line with this, meta-analysis of neuroscience literature using the mouse model as research subjects [2-4] has revealed that data collected from female mice (regardless of the estrous cycle) did not vary more than the data from males. In conclusion, we think that using randomized and mixed cohorts of male and female mice (as in the present study) would provide the same degree of variability in both sexes. Nevertheless, we have added a sentence to point to this possibility in the Discussion Section (Page 15, lines 32-37).

(2) The rationale for including parvalbumin (PV) cells in the study could be clarified. Is there prior evidence suggesting that this specific cell type is involved in mediated learning? This could apply to sensory stimuli not used in the current study.

In the revised version of the manuscript, we have better clarified why we targeted PV interneurons, specifically mentioning previous studies [5] (see Page 11, Lines 27-34).

(3) The photometry recordings from the dHPC during the preconditioning phase, shown in Figure 3, are presented as average responses. It would be beneficial to separate the early vs. late trials to examine whether there is an increase in hippocampal activity as the associative learning progresses, rather than reporting the averaged data. Additionally, to clarify the dynamics of the dHPC in associative learning, the authors could compare the magnitude of photometry responses when light and tone stimuli are presented individually in separate sessions versus when they are presented closely in time to facilitate associative learning.

As commented above, according to the Reviewer’s comment, we have now included a new Supplementary Figure 4, which splits the photometry data by the different preconditioning and conditioning sessions. Overall, this data suggests that there are no major changes on cell activity in both hippocampal regions during the different sessions as similar light-tone-induced enhancement of activity is observed. There is only an interesting trend in the activity of Pan-Neurons over the onset of light during conditioning sessions. All this is included now in the Results Section (Page 12, Line 13-15).

(4) The authors note that PV cell responses recorded with GCaMP were similar to general hippocampal neurons, yet chemogenetic manipulations of PV cells did not impact behavior. A more detailed discussion of this discrepancy would be helpful.

As suggested by the Reviewer, we have included additional Discussion to explain the potential discrepancy between the activity of PV interneurons assessed by photometry and its modulation by chemogenetics (see Page 16, Lines 27-33).

(5) All fiber photometry recordings were conducted in male mice. Given the sex differences observed in associative learning, the authors could expand the study to include dHPC responses in females during both preconditioning and conditioning sessions.

We appreciate the comment of the Reviewer. Indeed, thanks to other comments made by other Reviewers in this revision (see Point 1 of Reviewer #2), we are not still sure that we have an optimal protocol to study mediated learning in female mice due to sexspecific changes related to fear generalization. Thus, the revised version of the manuscript, although highlighting these sex differences in behavioral performance (see Supplementary Figure 2), is more focused in male mice and, accordingly, all photometry or chemogenetic experiments are performed exclusively using male mice. In future studies, once we would be sure to have a sensory preconditioning paradigm working in female mice, it will be very interesting to study if the same hippocampal mechanisms mediating this behavior in male mice are also observed in female mice.

Minor Comments:(1) In the right panel of Figure 2A, females received only one conditioning session, so the "x2" should be corrected to "x1" conditioning to accurately reflect the data.

We thank the Reviewer for the comment that has been addressed in the revised version of the manuscript.

(2) The overall presentation of Figure 3 could be improved. For example, the y-axis in Panel B could be cut to a maximum of 3 rather than 6, which would better highlight the response data. Alternatively, including heatmap representations of the z-score responses could enhance clarity and visual impact.

We thank the Reviewer for the comment that has been addressed providing a new format for Figures 2 and 3 in the revised version of the manuscript.

(3) There are several grammatical errors throughout the manuscript. It is recommended that the authors use a grammar correction tool to improve the overall writing quality and readability.

We have tried to correct the grammar through all the manuscript.

**Reviewer #2 (Recommendations for the authors):**
(1) In the abstract the authors write that sensory preconditioning requires the "repeated and simultaneous presentation of two low-salience stimuli such as a light and a tone". Previous research has shown that sensory preconditioning can still occur if the two stimuli are presented serially, rather than simultaneously. Further, the tone and the light are not necessarily "low-salience", for example, they can be loud or bright. It would be better to refer to them as innocuous.

In the revised version of the abstract, we have included the modifications suggested by the Reviewer.

(2) The authors develop a novel automated tool for assessing freezing behaviour in mice that correlates highly with both manual freezing and existing, open-source freeze estimation software (ezTrack). The authors should explain how the new program differs from ezTrack, or if it provides any added benefit over this existing software.

We have added new information in the Results Section (Page 10, Lines 13-20 to better explain how the new tool to quantify freezing could improve existing software).

(3) In Experiment 1, the authors report a sex difference in levels of freezing between male and female mice when they are only given one session of sensory preconditioning. This should be supported by a statistical comparison of levels of freezing between male and female mice.

Based on the new results obtained with female mice, we have decided to remove the original Figure 1 of the manuscript as it is not meaningful to compare male and female mediated learning response if we do not have an optimal protocol in female mice.

(4) Why did the authors choose to vary the duration of the stimuli across preconditioning, conditioning, and testing? During preconditioning, the light-tone compound was 30s, in conditioning the light was 10s, and at test both stimuli were presented continuously for 3 min. Did the level of freezing vary across the three-minute probe session? There is some evidence that rodents can learn the timing of stimuli and it may be the case that freezing was highest at the start of the test stimulus, when it most closely resembled the conditioned stimulus.

Differences in cue exposure between the preconditioning phase and the test phase were decided based on important differences between previous protocols used in rats compared to how mice are responding. Indeed, mice were not able to adapt their behavioral responses when shorter time windows exposing the cue were used as it clearly happens with rats1. In addition, we have added a new graph to show the time course of the behavioral responses (see Figure 1 and 4 and Supplementary Figure 2) that correlate with the quantification of freezing responses shown by the percentage of freezing during ON and OFF periods.

(5) The title of Experiment 1 "Female and male mice show mediated learning using an auditory-visual sensory preconditioning task" - this experiment does not demonstrate mediated learning; it merely shows that animals will freeze more in the presence of a stimulus as compared with no stimulus. This experiment lacks the necessary controls to claim mediated learning (which are presented in Experiment 2) and should therefore be retitled something more appropriate.

As stated above, based on the new results obtained with female mice, we have decided to remove the original Figure 1 of the manuscript as it is not meaningful to compare male and female mediated learning response if we do not have an optimal protocol in female mice.

(6) In Figure 2, why does the unpaired group show less freezing to the tone than the paired group given that the tone was directly paired with the shock in both groups?

We believe the Reviewer may have referred to the tone in error (i.e. there are no differences in the freezing observed to the tone) and (s)he might be talking about the freezing induced by the Light in the direct learning test. In this case, it is true that the direct learning (e.g. percentage of freezing) seems to be slightly lower in the unpaired group compared to the paired one, which could be due to a latent inhibition process caused by the different exposure of cues between paired and unpaired experimental groups. However, the direct learning in both groups is clear and significant and there are no significant differences between them, which makes difficult to extract any further conclusion.

(7) The stimuli in the design schematics are quite small and hard to see, they should be enlarged for clarity. The box plots also looked stretched and the colour difference between the on and off periods is difficult to discern.

We have included some important modification to the Figures in order to address the comments made by the Reviewer and improve its quality.

(8) The authors do not include labels for the experimental groups (paired, unpaired, no shock) in Figures 2B, 2D, 2C, and 2E. This made it very difficult to interpret the figure.

According to this suggestion, Figure 2 has been changed accordingly.

(9) The levels of freezing during conditioning should be presented for all experiments.

We have generated a new Supplementary Figure 9 to show the freezing levels during conditioning sessions.

(10) In the final experiment, the authors wrote that mice were injected with J60 or saline, but I could not find the data for the saline animals.

In the Results and Methods section, we have included a sentence to better explain this issue. In addition, we have added a new Supplementary Figure 7 to show the performance of all control groups.

(11) Please list the total number of animals (per group, per sex) for each experiment.

In the revised version of the manuscript, we have added this information in each Figure Legend.

**Reviewer #3 (Recommendations for the authors):**
I found this study very interesting, despite a few weaknesses. I have several minor comments to add, hoping that it would improve the manuscript:(1) The terminology used is not always appropriate/consistent. I would use "freely moving fiber photometry" or simply "fiber photometry" as calcium imaging conventionally refers to endoscopic or 2-photon calcium imaging.

We thank the Reviewer for this comment that has been addressed and corrected in the revised version of the manuscript.

(2) "Dorsal hippocampus mediates light-tone sensory preconditioning task in mice" suggests that a brain region mediates a task. I would rather suggest, e.g. "Dorsal hippocampus mediates light-tone association in mice"

We thank the Reviewer for this comment that has been addressed and corrected in the revised version of the manuscript.

(3) As you are using low-salience stimuli, it would be better to also inform the readership with the light intensity used for the light cue, for replicability purposes.

In the Methods section (Page 5, Line 30), we have added new information regarding the visual stimuli used.

(4) If the authors didn't use a background noise during the probe tests, the tone cue could have been perceived as being louder/clearer by mice. Couldn't it have inflated the freezing response for the tone cue?

This is an interesting comment made by the Reviewer although we do not have any data to directly answer his/her suggestion. However, the presence of the Background noise resulted necessary to set up the protocol and to change different aspects of the context through all the paradigm, which was necessary to avoid fear generalization in mice. In addition, as demonstrated before [6] , the presence of background noise is important to avoid that other auditory cue (i.e. tone) could induce fear responses by itself as the transition of noise to silence is a signal to danger for animals.

(5) "salience" is usually used for the intensity of a stimulus, not for an association or pairing. Rather, we usually refer to the strength of an association.

We thank the Reviewer for this comment that has been addressed and corrected in the revised version of the manuscript.

(6) Figure 3, panel A. "RCaMP Neurons", maybe "Pan-Neurons" would be more appropriate, as PV+ inter-neurons are also neurons.

We thank the Reviewer for this comment that has been corrected accordingly.

(7) Figure 4, panel A, please add the AAV injected, and the neurons labelled in your example slice.

We thank the Reviewer for this comment that has been corrected accordingly.

References

(1) Wong, F. S., Westbrook, R. F. & Holmes, N. M. 'Online' integration of sensory and fear memories in the rat medial temporal lobe. Elife 8 (2019). https://doi.org:10.7554/eLife.47085

(2) Prendergast, B. J., Onishi, K. G. & Zucker, I. Female mice liberated for inclusion in neuroscience and biomedical research. Neurosci Biobehav Rev 40, 1-5 (2014). https://doi.org:10.1016/j.neubiorev.2014.01.001

(3) Becker, J. B., Prendergast, B. J. & Liang, J. W. Female rats are not more variable than male rats: a meta-analysis of neuroscience studies. Biol Sex Differ 7, 34 (2016). https://doi.org:10.1186/s13293-016-0087-5

(4) Shansky, R. M. Are hormones a "female problem" for animal research? Science 364, 825-826 (2019). https://doi.org:10.1126/science.aaw7570

(5) Busquets-Garcia, A. et al. Hippocampal CB1 Receptors Control Incidental Associations. Neuron 99, 1247-1259 e1247 (2018). https://doi.org:10.1016/j.neuron.2018.08.014

(6) Pereira, A. G., Cruz, A., Lima, S. Q. & Moita, M. A. Silence resulting from the cessation of movement signals danger. Curr Biol 22, R627-628 (2012). https://doi.org:10.1016/j.cub.2012.06.015